# Tracing Ammonia Emission Sources in California's Salton Sea Region: Insights from Airborne Longwave-Infrared Hyperspectral Imaging and Ground Monitoring

Sina Hasheminassab[1*], David M. Tratt[2], Olga V. Kalashnikova[1], Clement S. Chang[2], Morad Alvarez[2], Kerry N. Buckland[2], Michael J. Garay[2], Francesca M. Hopkins[3], Eric R. Keim[2], Le Kuai[1], Yaning Miao[3], Payam Pakbin[4], William C. Porter[3], Mohammad H. Sowlat[4]

[1] Jet Propulsion Laboratory, California Institute of Technology, Pasadena, CA 91011, USA
[2] The Aerospace Corporation, Los Angeles, CA 90009, USA
[3] University of California, Riverside, CA 92521, USA
[4] South Coast Air Quality Management District, Diamond Bar, CA 91765, USA

* Correspondence to: Sina Hasheminassab (sina.hasheminassab@jpl.nasa.gov)

## Abstract

Ammonia ($NH_3$) plays an important role in atmospheric chemistry and air quality, but its emissions remain poorly constrained due to its short atmospheric lifetime, high spatial heterogeneity, and limited coverage of existing monitoring resources. This study integrates airborne longwave-infrared (LWIR) hyperspectral imaging at ~2 m spatial resolution with ground-based stationary and mobile in-situ measurements to map and characterize $NH_3$ emissions in two regions near the Salton Sea in Southern California: Mecca in the northwest and Imperial in the southeast. Airborne surveys conducted in March and September 2023 with a wide-swath LWIR spectral imager revealed pronounced spatial and seasonal variability. Average $NH_3$ levels in Imperial were 2.5 to 8 times higher than those in Mecca, linked primarily to large, concentrated animal feeding operations (CAFOs), geothermal power plants, fumaroles, and intensive agricultural activities. Ground-based mobile monitoring corroborated these findings, showing elevated $NH_3$ levels near these sources and especially high $NH_3$ concentrations downwind of CAFOs with large cattle populations. The results underscore the utility of airborne LWIR hyperspectral imaging in detecting and mapping $NH_3$ at hyperlocal scales, including sources absent from existing inventories. They further highlight the need for routine airborne campaigns and the development of next-generation satellite missions with higher spatial resolution to achieve comprehensive, large-area monitoring. These findings inform air quality management strategies and emphasize the importance of improving emission inventories for effective mitigation of $NH_3$-driven air pollution.

## 1. Introduction

Ammonia ($NH_3$) is a critical component of the atmospheric nitrogen cycle and an important precursor for the formation of fine particulate matter ($PM_{2.5}$) (Tang et al., 2025; Wyer et al., 2022a). It is primarily emitted from agricultural activities, particularly livestock waste and the application of fertilizers, but also from combustion sources such as vehicles and biomass burning (Beaudor et al., 2025; Vira et al., 2022; Zeng et al., 2018). As the

primary form of reactive nitrogen in the environment, it plays an important role in nitrogen deposition,
eutrophication, and acidification processes (Li et al., 2025; Zhang et al., 2012). Additionally, atmospheric $NH_3$ reacts
with acidic compounds like nitric and sulfuric acids to form ammonium salts, contributing significantly to secondary
$PM_{2.5}$, leading to various adverse health outcomes (Wyer et al., 2022b).
Recent studies have reported increasing trends in atmospheric $NH_3$ levels at global and regional scales, driven
largely by agricultural activities, industrial processes, and changing climatic conditions (Kuttippurath et al., 2024;
Ma et al., 2025; Van Damme et al., 2021; Wang et al., 2023; Yu et al., 2018). For example, Van Damme et al. (2021)
reported a 12.8 % global increase between 2008 and 2021 based on the Infrared Atmospheric Sounding
Interferometer (IASI) satellite measurements. During the same period, also using IASI data, Wang et al. (2023)
observed a 6.8 % annual increase in $NH_3$ concentrations across the contiguous U.S., with hotspots concentrated in
agricultural regions. Also, during the 2020 COVID-19 lockdown, Kuttippurath et al. (2024) found that unlike most
pollutants which declined, $NH_3$ levels rose, largely due to sustained or increased agricultural activity. Consequently,
regulatory bodies such as the European Parliament and air quality agencies in California are recognizing $NH_3$ as an
emerging pollutant of concern (European Commission, 2022; South Coast AQMD, 2024). Furthermore, the recent
tightening of the U.S. Environmental Protection Agency's (EPA) National Ambient Air Quality Standards (NAAQS)
for annual average $PM_{2.5}$ to 9 µg m$^{-3}$ has drawn increasing attention to $NH_3$, particularly in non-attainment areas, to
better characterize its sources and contribution to secondary $PM_{2.5}$ (Reconsideration of the National Ambient Air
Quality Standards for Particulate Matter, 88 Fed. Reg. 5558 | Casetext, 2025). However, unlike other key gas-phase
precursors of $PM_{2.5}$ such as $NO_2$ and $SO_2$, which are regulated as criteria pollutants in the U.S. and are widely
monitored, the scarce monitoring of $NH_3$ and uncertainties in emissions inventory hinder accurate assessments of its
contribution to secondary $PM_{2.5}$ formation.
Despite its importance, measuring and characterizing $NH_3$ emissions are challenging due in part to its short
atmospheric lifetime, high spatial heterogeneity, reactivity, and solubility (Nair and Yu, 2020). Established ground-
based monitoring networks, such as the Ammonia Monitoring Network (AMoN) in the U.S., provide high-quality
data but are spatially sparse and temporally limited to biweekly measurements (Felix et al., 2017). Existing satellite
instruments capable of detecting $NH_3$ (e.g., CrIS, IASI, AIRS) offer broader spatial coverage, but generally at coarse
spatial resolutions (15-50 km) inadequate for the identification of local-scale emission sources (Dammers et al.,
2019; Shephard et al., 2020; Warner et al., 2016). Due to these challenges, current chemical transport model
emission inventories often fail to adequately represent major $NH_3$ sources, leading to significant underestimation of
their contributions (Burns et al., 2023).
The longwave-infrared (LWIR) hyperspectral imaging technique is highly effective at identifying diagnostic spectral
features of various environmentally important gases, with a particularly strong sensitivity to $NH_3$ (Kuai et al., 2019;
Leifer et al., 2017). Airborne deployment of this technique enables wide-area, high-resolution mapping of $NH_3$
emissions. Airborne instruments such as The Aerospace Corporation's Mako imager and JPL's Hyperspectral
Thermal Emission Spectrometer (HyTES) have demonstrated the capability to detect and quantify $NH_3$ with spatial
resolutions ranging from 1 to 30 m, depending on flight altitude (Hall et al., 2011; Hulley et al., 2016; Kuai et al.,
2019; Tratt et al., 2013). This high-resolution capability is particularly important in complex regions where multiple
emission sources coexist.
The Salton Sea region in Southern California represents an important $NH_3$ hotspot with unique air quality challenges
and opportunities for study. Intensive agricultural activities, including large-scale concentrated animal feeding
operations (CAFOs) and extensive fertilizer application, as well as industrial and natural geothermal emissions,
contribute to its high $NH_3$ levels. The Imperial Valley, located in the southern part of the Salton Sea, has one of the
highest densities of CAFOs in California and ranks among the top ten $NH_3$ hotspots in the U.S. (Wang et al., 2023).
Despite its significance, $NH_3$ sources in this region have not been extensively monitored or thoroughly
characterized.
This study aims to address these gaps by integrating airborne LWIR hyperspectral imaging with ground-based
stationary and mobile in-situ monitoring to map and characterize $NH_3$ emissions in the Salton Sea region. Using this
multi-platform approach, we identify major $NH_3$ sources, including previously undocumented or underrepresented
ones, and assess their spatial and temporal variability in these complex environments. The findings provide a
foundation for improving $NH_3$ inventories and underscore the necessity for routine airborne measurements and
development of satellite missions with higher spatial resolution to resolve persistent data gaps in understanding
$NH_3$'s contribution to atmospheric chemistry and air quality. This demonstration is timely, given the growing global
demand for better $NH_3$ monitoring at scale.
**2. Methodology**
**2.1. Overview**
The study focused on the Salton Sea region in Southern California, selected for its diverse and significant $NH_3$
emission sources and unique environmental characteristics. Measurements included airborne remote sensing and
ground-based in-situ monitoring, both mobile and stationary. Figure 1 illustrates the study areas, the footprints of
airborne remote sensing measurements, the routes covered by the mobile platform, and the location of the air
monitoring site utilized in this research. The following sections detail each component of the study.

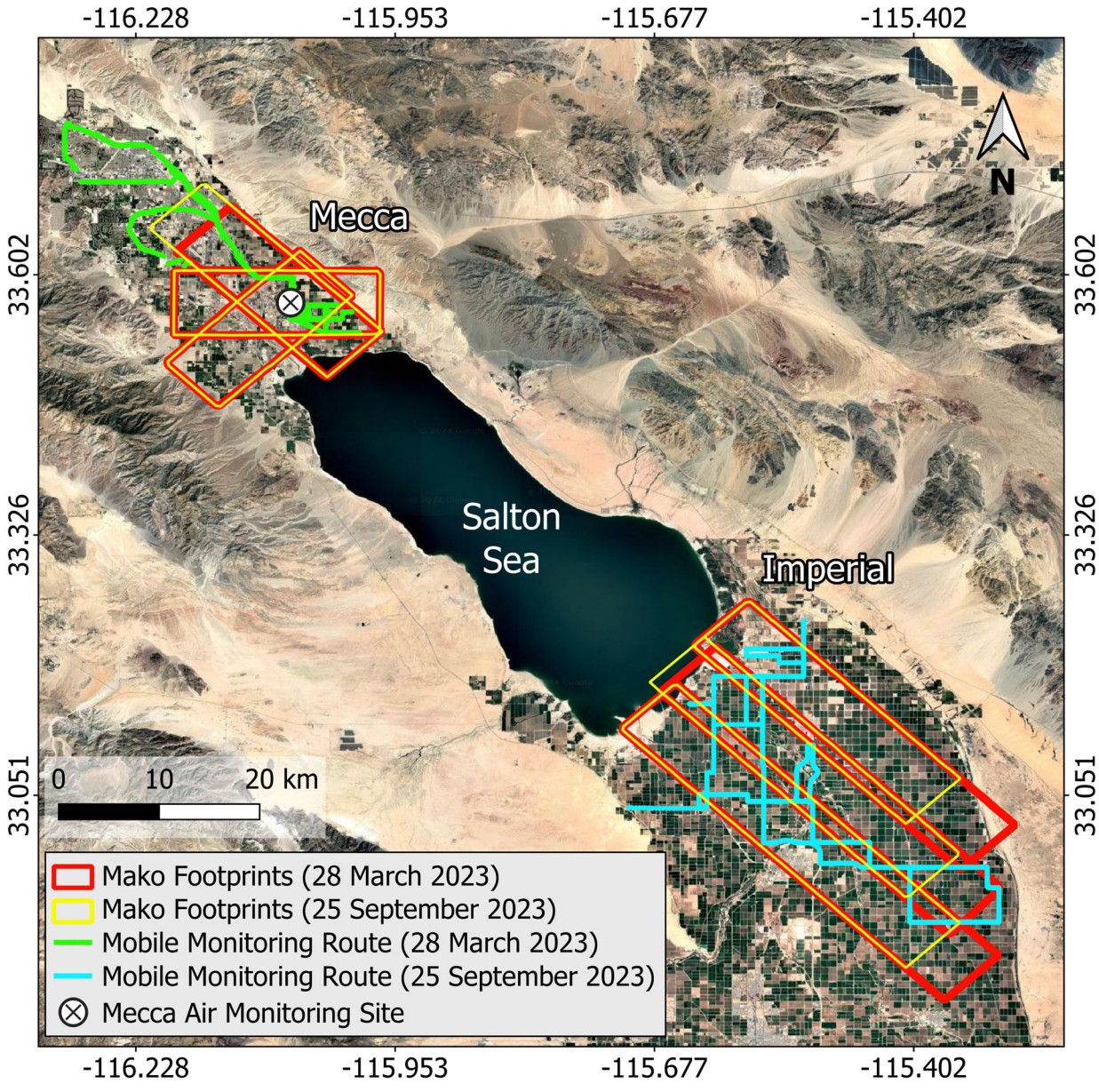

**Figure 1. Map of study areas around the Salton Sea. The northwestern area is denoted as "Mecca," and the southeastern area as "Imperial," and they are referred to as such throughout the paper. Base imagery ©2025 Google.**

### 2.2. Study Areas

The study areas encompass the northwest and southeast regions of the Salton Sea, hereafter referred to as Mecca and Imperial, respectively (Figure 1). Both areas are predominantly rural and surrounded by extensive agricultural lands. Mecca is known for cultivating high-value crops such as dates, citrus fruits, and vegetables, while Imperial is characterized by large-scale agricultural production, with a focus on crops including alfalfa, grains, and vegetables. Additionally, Imperial has a high concentration of CAFOs and several geothermal power plants (GPPs). Both regions face severe air pollution challenges, including agricultural emissions and windblown dust from surrounding desert regions as well as from the increasingly exposed dried lakebed surfaces (Bi et al., 2020; Frie et al., 2019;

Johnston et al., 2019; Lieb et al., 2024). Previous studies have reported a high prevalence of health issues related to
air pollution in this area, particularly impacting cardiovascular and respiratory systems (Farzan et al., 2019; Johnston
et al., 2024; Jones and Fleck, 2020; Miao et al., 2025). Both regions are designated as nonattainment for 8-hour
ozone and 24-hour $PM_{10}$ standards, and Imperial is additionally nonattainment for annual $PM_{2.5}$ (Current
Nonattainment Counties for All Criteria Pollutants, 2025). Moreover, according to the California Office of
Environmental Health Hazard Assessment (OEHHA), these regions rank among the communities with the highest
pollution burdens and socioeconomic disadvantages in California and are therefore classified as environmental
justice communities.

### 2.3. Airborne Measurements

The airborne component of the study utilized The Aerospace Corporation's Mako imager, mounted on a Twin Otter
aircraft. Mako is a 3-axis stabilized whiskbroom imager spanning the LWIR spectral region (specifically 7.6-13.2
μm) in 128 contiguous channels capable of detecting and quantifying multiple gas species. The performance and
operating specifications of Mako have been detailed elsewhere (Buckland et al., 2017; Hall et al., 2016). For this
study, flights were conducted at an altitude of approximately 3,750 m above ground level (AGL), with a 7.5-
kilometer swath and a nominal ground sample distance (GSD) of 2.1 m. Flight measurements were conducted on
two different days in 2023, during distinct seasons: March 28th and September 25th. Figure 2 presents the airborne
data acquisition times for each day and region. On March 28th, measurements began over Mecca at approximately
10:37 AM PDT and continued until 11:08 AM PDT, followed by acquisitions in the Imperial region from 11:24 AM
to 12:12 PM PDT. On the return to base, another measurement was conducted over the SE-to-NW flight line over
Mecca (see Figure 4) between 12:40 PM and 12:46 PM PDT. Drawing from the experience gained during the March
campaign, the September flights aimed to further differentiate morning and afternoon measurements. Thus, on
September 25th, flight measurements commenced earlier, starting over Mecca at 9:35 AM PDT and continuing until
10:10 AM PDT. Data acquisition in the Imperial region was then conducted from 10:27 AM to 11:12 AM PDT. A
second round of measurements, over the same flightlines, was performed in the afternoon, with acquisitions over
Mecca between 3:06 PM and 3:39 PM PDT and over Imperial from 3:53 PM to 4:37 PM PDT. The flight paths were
identical on both days, with the exception of a northwestern extension of one flight line over Mecca and slightly
shorter flight lines over Imperial on September 25th, as shown in Figure 1. The extension over Mecca was intended
to capture the source of a plume attributable to a waste recycling facility that had been partially detected during the
March flights. More details about the emissions from this facility are provided in the results section. The shortened
flight lines in September were to avoid approaching restricted airspace that was encountered during the March
flights.

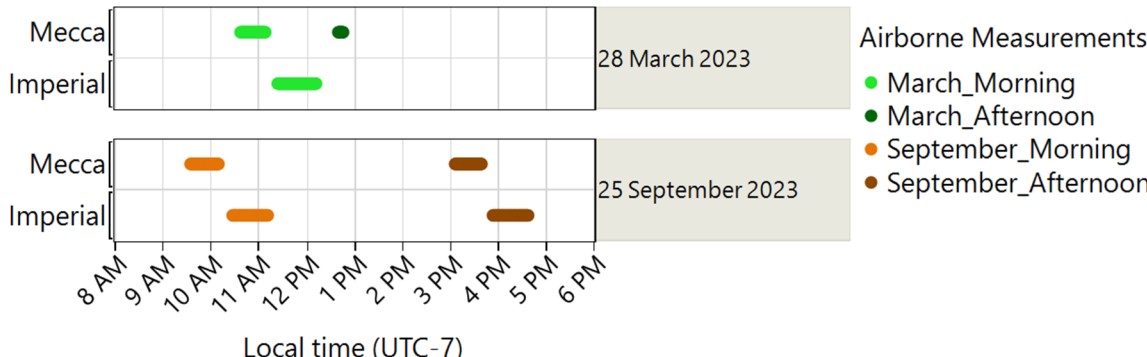

**142**

**143** **Figure 2. Timing of airborne measurements over the Mecca and Imperial study areas on March 28 and September 25,**
**144** **2023. The time periods defined in the legend are used throughout the paper to distinguish measurement sessions.**

**145** During the September campaign, an additional series of airborne measurements in multi-scan (movie) mode was

**146** conducted over nine pre-determined sources, including GPPs, fumaroles, CAFOs, and a waste recycling facility.

**147** These measurements, collected at a nominal frame rate of 0.6 Hz during both morning and afternoon flight sessions,

**148** captured short-term temporal variations in $NH_3$ emissions from these sources. The data will be used to estimate

**149** emission fluxes from these sources, which will be the subject of a separate publication from this study.

**150** ### 2.4. $NH_3$ Retrieval

**151** Details of the $NH_3$ retrieval methodology from Mako are described in Hall et al. (2025) and Tratt et al. (2025).

**152** Briefly, $NH_3$ column densities were retrieved using a custom Scene-based Algorithm for Gas Estimation (SAGE),

**153** which employs a nonlinear radiative transfer model and auxiliary data to estimate gas concentrations along the

**154** atmospheric path, reported in units of ppm-m (parts per million-meter). To evaluate the confidence of each retrieval,

**155** SAGE calculates a t-statistic that constitutes a measure of the retrieval signal-to-noise ratio. The algorithm also

**156** outputs fit residuals and the thermal contrast ($\Delta T$), defined as the temperature difference between the near-surface

**157** air parcel and the underlying surface, for each pixel. In this study, pixels were thresholded using a t-statistic of $\geq 3.0$

**158** and $|\Delta T| \geq 2.0$ °C to eliminate high-uncertainty $NH_3$ abundance estimates. It should be noted that in all $NH_3$ column

**159** density maps in this paper, null pixels that did not meet the t-statistic and $\Delta T$ thresholds are rendered transparent to

**160** expose the underlying imagery.

**161** The retrievals in this study were also validated against ground-based in-situ $NH_3$ measurements by incorporating air

**162** temperature and planetary boundary layer height (PBLH) fields from NOAA's High-Resolution Rapid Refresh

**163** (HRRR) model and Gaussian plume modeling to characterize atmospheric conditions and plume dynamics. The

**164** minimum detectable concentration (MDC) of ground-level $NH_3$ depends on thermal contrast and boundary layer

**165** height, and ranged from 3 to 33 ppbv over Mecca in September under varying atmospheric conditions. Validation

**166** against ground-based in situ measurements showed good agreement, with differences ranging from 16% to 37%

**167** under stable and well-mixed conditions. Detailed descriptions of the retrieval process, detection sensitivity, and

**168** validation results for the airborne component of this study are provided in Tratt et al. (2025).

### 2.5. Ground-Based In-Situ Stationary Monitoring

Continuous 1-minute $NH_3$ monitoring was conducted using a Picarro G2301 analyzer at the South Coast Air Quality Management District (South Coast AQMD) Mecca air monitoring site (AMS) (Figure 1). Routine quality assurance procedures, including monthly zero and span checks, were conducted by South Coast AQMD. Unfortunately, the instrument was non-operational for the entire month of March 2023, concurrent with the first day of airborne measurements. However, it was operational before and after March 2023, allowing the assessment of $NH_3$ levels on the second airborne measurement day in September and evaluation of its temporal trends throughout 2023.

### 2.6. Ground-Based In-Situ Mobile Monitoring

The mobile monitoring was conducted using a mobile platform developed and operated by the University of California, Riverside. This mobile platform was equipped with advanced instrumentation to continuously measure geolocated atmospheric $NH_3$ levels along roadways. The primary sensor was a Picarro G2123 cavity ring-down spectrometer, measuring $NH_3$ concentrations every ~3 s with a precision of 500 ppt, sampling ambient air through a ¼" Teflon line at a rate of 1.7 dm³ min⁻¹ from an inlet positioned 3 m AGL. GPS data, recorded every second by a roof-mounted Garmin GPS 16X receiver, provided precise geographical coordinates.

Mobile monitoring was conducted concurrently with the airborne measurements, with the routes shown in Figure 1. On March 28th, during the first airborne campaign and in the absence of stationary monitoring at the Mecca AMS, the mobile platform focused on the Mecca region. On September 25th, during the second airborne campaign, mobile monitoring covered the Imperial region.

### 2.7. Meteorology

Meteorological data, including temperature, wind speed, and direction at 10 m AGL, and PBLH, corresponding to the study areas and flight measurement periods, were retrieved from NOAA's HRRR model output using the Herbie Python package (https://zenodo.org/records/13329302; accessed 4 January 2025). Figures S1 and S2 provide a summary of these meteorological parameters across the study regions during each flight measurement session.

Temperatures were generally comparable between Mecca and Imperial, ranging from approximately 20°C in March to 31°C during the September morning flights, and rising further to 37°C in September afternoon. During the March campaign, PBLH values ranged from 730 to 937 m across both regions. During the September morning flights, which began earlier in the day, PBLH values were lower, around 340 m in Mecca and 554 m in Imperial. However, by the afternoon, PBLH significantly increased to 1,821 m in Mecca and 1,592 m in Imperial.

The wind direction/speed varied largely across the study area and over time. During the March morning flights, both Mecca and Imperial experienced light winds ranging from 0 to 2 m s⁻¹, with Mecca dominated by southerly winds and Imperial by northwesterly winds. By the afternoon, wind speeds in Mecca increased substantially, exceeding 4 m s⁻¹. In September morning, winds were predominantly northwesterly in both regions, with Mecca exhibiting stronger winds averaging 1.6 m s⁻¹ compared to 0.95 m s⁻¹ in Imperial. By the afternoon, wind speeds increased in

both areas; while Imperial's winds remained primarily northwesterly with an average speed of 1.5 m s$^{-1}$, Mecca
experienced a notable shift in wind direction to southeasterly, with an average speed of 3.1 m s$^{-1}$.

**2.8. Spatial Data of Potential NH$_3$ Emitters**

The study area around the Salton Sea is home to several notable NH$_3$ emission sources, including fumaroles,
CAFOs, GPPs, and waste recycling facilities. To attribute the observed NH$_3$ hotspots to these known emitters—and
potentially identify any previously unknown or unreported sources—geospatial data on their locations and attributes
were compiled from various sources. Data on GPPs, including their locations and generating capacities, were
obtained from the California State Geoportal (https://gis.data.ca.gov, accessed January 4, 2025). Information on
CAFOs, such as facility locations and cattle populations, was sourced from the California Air Resources Board's
(CARB) California Dairy & Livestock Database (CADD) (https://ww2.arb.ca.gov/resources/documents/california-
dairy-livestock-database-cadd, accessed January 4, 2025). Information about waste recycling facilities was obtained
from EPA's Toxic Release Inventory (TRI) (https://www.epa.gov/toxics-release-inventory-tri-program/tri-toolbox,
accessed January 4, 2025). Additionally, the locations of known fumaroles along the southern shore of the Salton
Sea were gathered from previous studies conducted in the region (Adams et al., 2017; Tratt et al., 2011, 2016).
Nearly all of the abovementioned sources were located in the Imperial study area, with the exception of a waste
recycling facility located in the Mecca region.
Figure S3 shows the total cattle population at CAFO facilities within the Imperial study area in 2022. Among these,
CAFO 1 stands out with the largest cattle population, approximately 150,000, far exceeding the numbers at other
CAFOs in the study area. In fact, according to CADD, CAFO 1 has the largest cattle population of any CAFO in
California (data not shown). Additionally, CAFOs 2 and 9 rank seventh and eighth statewide, respectively. Figure S4
shows the generating capacities of GPPs in the Imperial region, with GPPs 9, 3, 4, 5, and 11 having the highest
capacities, ranging from 50 to 55 MW.

**2.9. Data Analysis**

Data analyses, visualizations, and mapping were performed using R, QGIS, and SAS JMP. To investigate the source
directionality of NH$_3$ at the Mecca AMS, bivariate directional analyses were conducted using the openair package in
R.

**3. Results**

**3.1. Airborne Measurements**

Figure 3 shows the average NH$_3$ column density across both study areas during each airborne measurement session.
As shown, average NH$_3$ levels were significantly higher in Imperial compared to Mecca, with respective factors of
approximately 8 and 2.5 during the March and September campaigns. Seasonally, Mecca NH$_3$ levels were about 3.5
times higher in September than in March, while no significant seasonal variation was observed over Imperial.
Additionally, during the September campaign, afternoon NH$_3$ levels were 20–50 % higher than those measured in
the morning across both study areas.

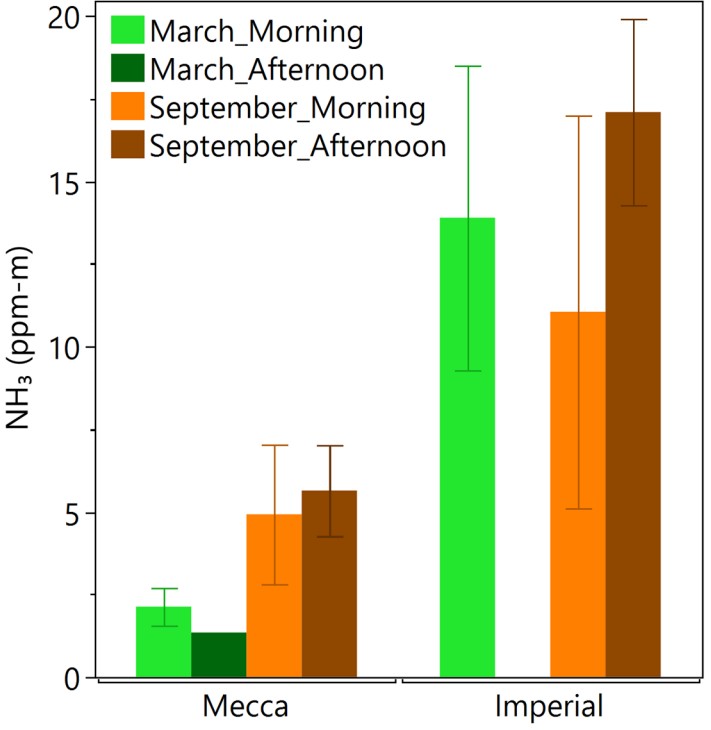

**Figure 3. Average NH₃ column density (ppm-m) for each study area and airborne measurement session. Error bars represent one standard deviation across measurements from different flight lines in each study area.**

Figures 4 and 5 illustrate NH₃ column density maps over Mecca and Imperial, respectively, from each airborne measurement session. Over Mecca in March, a distinct, strong plume was observed in the northwest part of the study area during the morning but dissipated by the afternoon. A closer examination of this plume in Figure S5 suggests it likely originated from a waste recycling facility located upwind, which was not fully captured during the March flight acquisition. As previously noted, in September, the flight measurements were extended northwestward to encompass this facility. The results showed detectable but relatively weaker plumes during both the morning and afternoon measurements. According to EPA TRI, this facility emitted over 27,500 pounds (12,470 kg) of NH₃ in 2023. Another notable observation over Mecca is the significant increase in NH₃ column density during the September afternoon measurements compared to the morning. The majority of hotspots were diffuse emissions concentrated over agricultural fields. This increase is likely due to higher afternoon temperatures, which enhance NH₃ volatilization from soil and plant surfaces, particularly in areas where fertilizers have been recently applied. Another possible explanation is the higher thermal contrast during the afternoon flights, which enhances the Mako sensor's detection sensitivity and enables the identification of NH₃ signals even at relatively low concentrations. Additional information about retrieval sensitivity can be found in the discussion section.

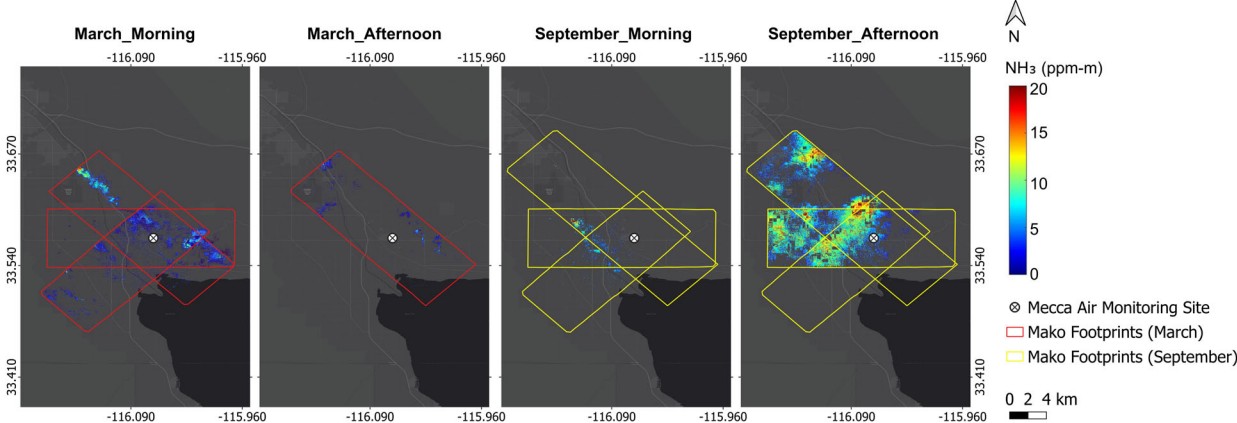

**Figure 4. Retrieved NH$_3$ column density (ppm-m) maps over the Mecca study area during each airborne measurement session. Null pixels that did not meet the t-statistic and ΔT thresholds are rendered transparent. Map sources: Esri, HERE, Garmin, INCREMENT P, ©OpenStreetMap contributors, and the GIS User Community.**

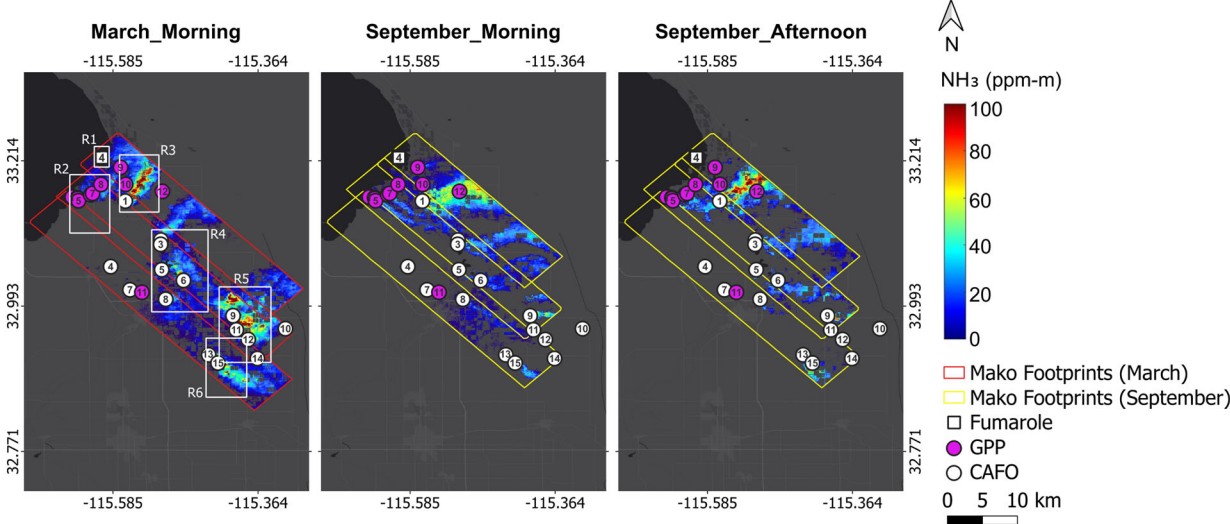

**Figure 5. Retrieved NH$_3$ column density (ppm-m) maps over the Imperial study area during each airborne measurement session. Null pixels that did not meet the t-statistic and ΔT thresholds are rendered transparent. The markers in each panel denote known NH$_3$ emission sources. Six hotspot regions, shown with white boxes, are defined for further investigation. Map sources: Esri, HERE, Garmin, INCREMENT P, ©OpenStreetMap contributors, and the GIS User Community.**

Significantly higher NH$_3$ levels were measured in the Imperial region. To properly illustrate the spatial gradient, the column density scale was adjusted from 0–20 ppm-m (used for the Mecca area) to 0–100 ppm-m. Multiple hotspots were detected in Imperial, and to identify the emitters, the locations of known point sources, including fumaroles, CAFO facilities, and GPPs, were overlaid on the maps (Figure 5). A version of these maps without the markers is included in the Supplementary Information Figure S6, to provide a clearer depiction of the detected plume patterns. Due to the complexity of this region, with several localized hotspots and sources in close proximity, six regions were

identified for further investigation (Figure 5). Moreover, Figure 6 depicts the pixel-level NH$_3$ column density
distributions in each region and flight measurement session.

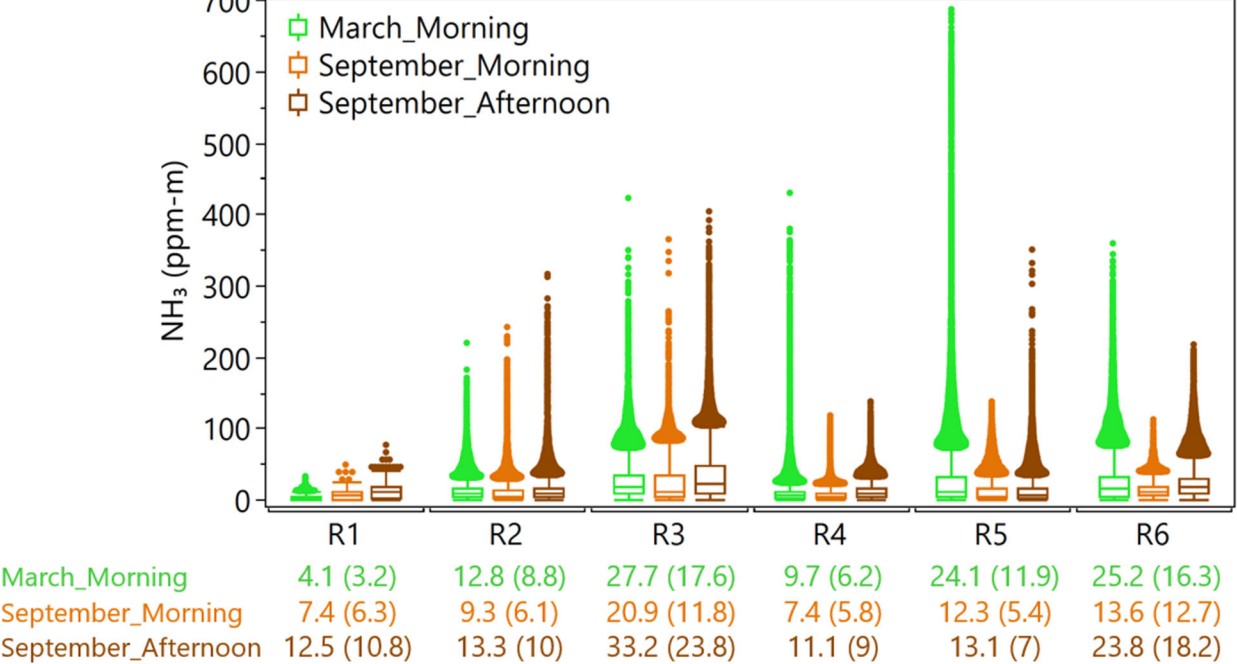


**Figure 6. Box plots of pixel-level retrieved NH$_3$ column density (ppm-m) for each predefined region and measurement**
**session over the Imperial study area. Boxes represent the interquartile range (IQR), with the horizontal line inside each**
**box indicating the median value. Whiskers extend to 1.5 times the IQR, and outliers are values exceeding 1.5 times the**
**IQR. Arithmetic mean (median) values are reported below the plot.**

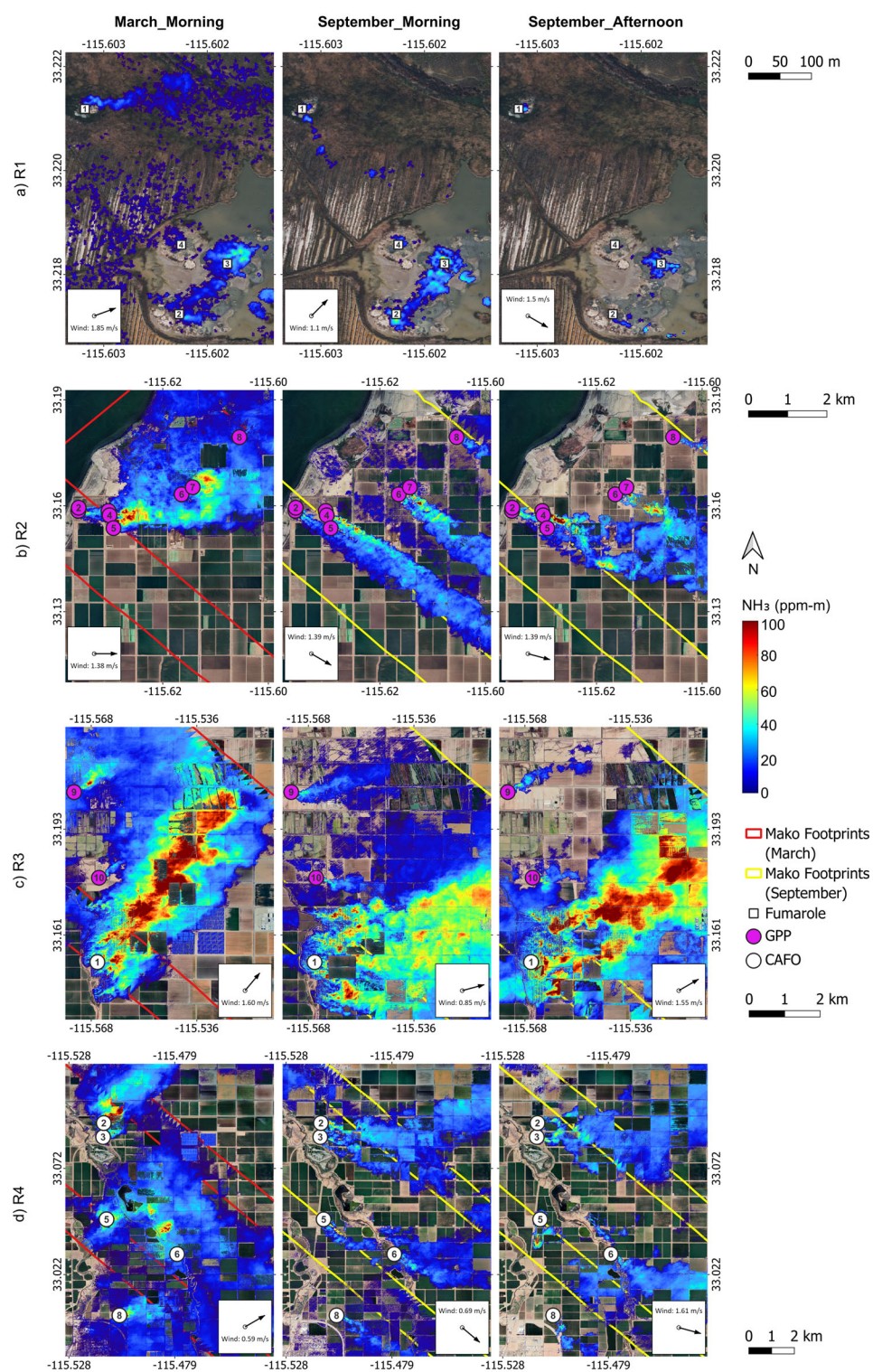


**Figure 7 (a-d). Retrieved NH₃ column density (ppm-m) maps over predefined Regions 1–4 in the Imperial study area**
**during each airborne measurement session. Null pixels that did not meet the t-statistic and ΔT thresholds are rendered**
**transparent. The markers in each panel denote known NH₃ emission sources. Wind directions were inferred from plume**
**patterns, and average wind speeds were calculated from the nearest HRRR grid cells. Base imagery ©2025 Google.**
Region 1 (Figure 7a) encompasses a number of known fumaroles along the southeastern shoreline of the Salton Sea
that appear to be associated with the Calipatria Fault (Lynch and Hudnut, 2008). Measurements over this area
revealed fluctuating $NH_3$ emissions from these natural geothermal vents, though the levels were not as substantial as
other sources detected in the region. Across the three airborne acquisitions over this region, average $NH_3$ column
density ranged between 4.1 and 12.5 ppm-m. Fumaroles, associated with geothermal activity, emit $NH_3$ as
superheated water and gases interact with nitrogen-bearing, organic-rich sediments. This process triggers chemical
reactions and the thermal decomposition of organic matter, releasing $NH_3$ and other gases into the atmosphere (Tratt
et al., 2011). These emissions are inherently variable, influenced by factors such as the intensity of geothermal
activity, sediment composition, and local geological conditions (Tratt et al., 2016). The detection and quantification
of these emissions were made possible by the high-spatial resolution (~2 m) of LWIR imaging, which allowed for
the identification of emissions at a fine scale that would otherwise be undetectable.
Region 2 (Figure 7b) comprises eight known GPPs, with generating capacities ranging from 10 to 54 MW. Airborne
measurements detected $NH_3$ emissions from all of these facilities, consistently across both seasons and at different
times of the day. In the September morning measurements, the dispersion and overlap of $NH_3$ emissions from this
cluster of GPPs tracked to a distance of approximately 7.5 km downwind. Moreover, the use of high-spatial
resolution measurements allowed for precise localization of emission points within each facility. In nearly all plants,
$NH_3$ emissions were primarily concentrated around the cooling towers. Figure S7 shows zoomed-in views of
emissions over GPPs 6, 7, and 9, illustrating the highest $NH_3$ emissions from their cooling towers. This pattern is
consistent with standard operational procedures in geothermal plants, where $NH_3$ is separated from the geothermal
brine and vented through the cooling tower exhaust. This method of disposal is common in such facilities as it
enhances the overall efficiency of the power plant while preventing $NH_3$ from causing operational issues, such as
corrosion (Frick et al., 2019; Nogara and Zarrouk, 2018).
Region 3 (Figure 7c), which contains the largest CAFO and the highest-capacity GPP in the study area, consistently
recorded the highest average $NH_3$ column density among all regions during each flight session, ranging from 20.9 to
33.2 ppm-m, with maximum pixel-level values exceeding 350 ppm-m. The emissions from CAFO 1 were
remarkably large, with plumes that far outsized those of any other sources in the entire study area. Across the three
flight acquisitions over Region 3, the width of the plumes from CAFO 1 ranged from 2.2 to 2.8 km and extended 7.1
to 7.7 km downwind of the source. In comparison, plume widths and lengths from the two nearby GPPs in this
region ranged from 0.1 to 0.3 km and 1 to 2.7 km, respectively. CAFOs emit $NH_3$ primarily due to the breakdown of
nitrogen-rich compounds in livestock manure, particularly urea in urine. Urea undergoes enzymatic hydrolysis when
it comes into contact with fecal matter, producing $NH_3$ gas. Factors such as manure management, ambient
temperature, pH, and wind speed influence this process (Hiranuma et al., 2010; Hristov et al., 2011). As previously
noted, CAFO 1 has the largest cattle population in the state, which explains the substantial $NH_3$ emissions from this
facility.
Regions 4, 5, and 6 (Figures 7d and S8 a-b) each contain multiple CAFOs, with detectable emissions from every
facility during each flight measurement, though the magnitudes varied among facilities. Region 6, encompassing

CAFOs 13 and 15, exhibited the second-highest average $NH_3$ column densities during each flight measurement, with levels ranging from 13.6 to 25.2 ppm-m. CAFOs 2 and 9, which have the second and third largest cattle populations (Figure S3), displayed particularly high $NH_3$ emissions during the March morning measurements, with maximum pixel-level column densities reaching 400 and 700 ppm-m, respectively.

Closer examination of the $NH_3$ column density maps for Regions 4 and 5 revealed emissions from a few facilities not included in the known emission sources overlaid on the maps. Specifically, in Region 4, two detectable emission sources were identified north of CAFO 6. Similarly, in Region 5, a notable emission source was detected north of CAFO 9. Figure S9 provides a zoomed-in view of these sources, which appear to be CAFOs based on satellite imagery but were not listed in the CADD database. Notably, the emissions from the non-inventoried source north of CAFO 9 were particularly significant during the March morning measurements, with pixel-level $NH_3$ column densities far exceeding 100 ppm-m.

Overall, the airborne measurements demonstrated a strong capability in identifying $NH_3$ emission sources. Quantitatively, the surveys detected $NH_3$ emissions from all four known fumarole geothermal vents, all 12 GPPs, and all 13 known CAFOs within the study area. Additionally, the high spatial resolution enabled the identification of at least three previously uninventoried point emission sources, appearing as CAFOs based on satellite imagery but not listed in existing databases, highlighting the method's ability to discover undocumented emitters.

### 3.2. Ground-Based In-Situ Mobile Monitoring

Mobile $NH_3$ monitoring was conducted primarily between 8:00 AM and 6:00 PM, concurrent with airborne measurements over Mecca (March 28th) and Imperial (September 25th). Figures 8 and 1 illustrate the time series of ground-level $NH_3$ concentrations on each monitoring day and map of the areas covered by the mobile platform, respectively. Gaps in the time series plots (Figure 8) correspond to periods when the instrument operated outside its nominal temperature and flow rate ranges, resulting in unrecorded concentration measurements. $NH_3$ concentrations were significantly higher in Imperial compared to Mecca, with average levels of 23.8 ppb in Mecca on March 28th and 250.1 ppb in Imperial on September 25th. The time series plots reveal multiple distinct plumes in the Imperial study area, detected as the mobile platform passed various sources, while the Mecca measurements indicated only a single notable increase above the background levels. To pinpoint the locations and potential emission sources, individual plumes were first identified based on distinct peaks in the time series plot (Figure 8), and their corresponding spatial locations were then mapped in Figure 9. Additionally, Figure 10 compares $NH_3$ concentration distributions across the identified plumes and highlights nearby known sources as potential emission contributors. In interpreting the mobile monitoring data, it is important to consider that the recorded $NH_3$ levels reflect concentrations at the sampling height (~3m AGL) and, thus, any plumes passing above this height may not have been fully captured by the mobile platform.

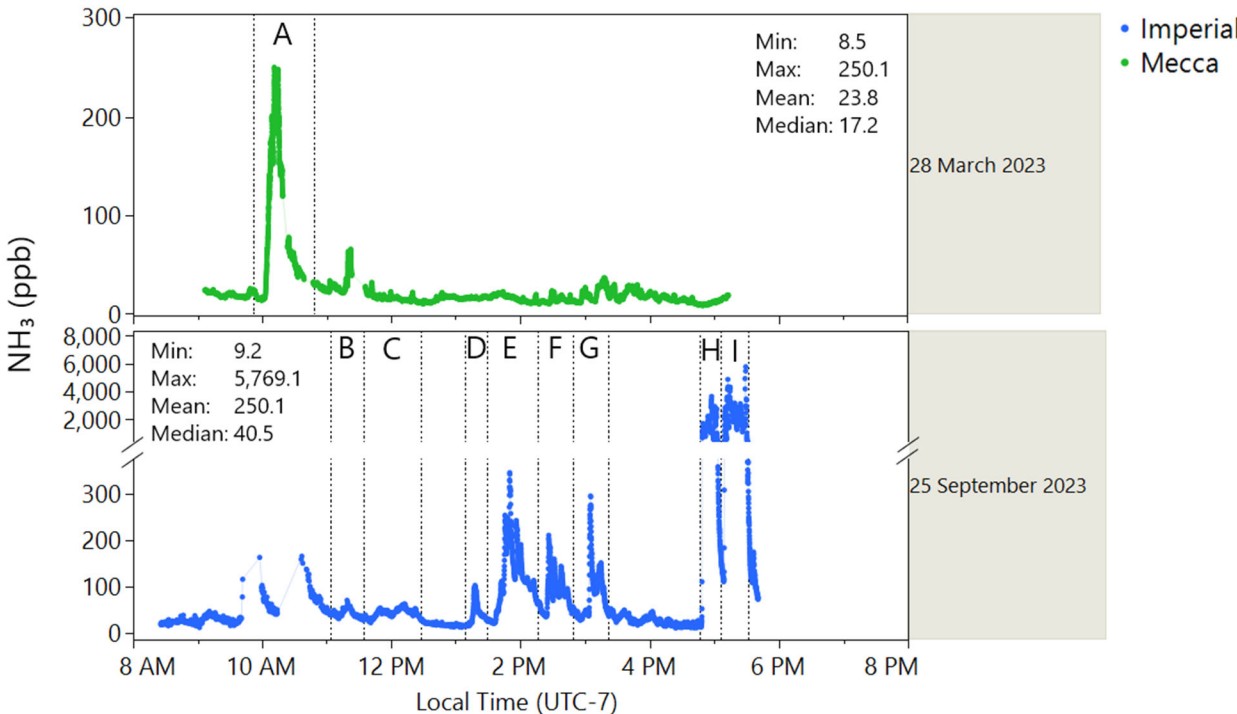


**Figure 8. Time series of instantaneous ground-level NH₃ concentrations (ppb) measured by the mobile platform on each measurement day, with individual plumes labeled A to I for further investigation.**


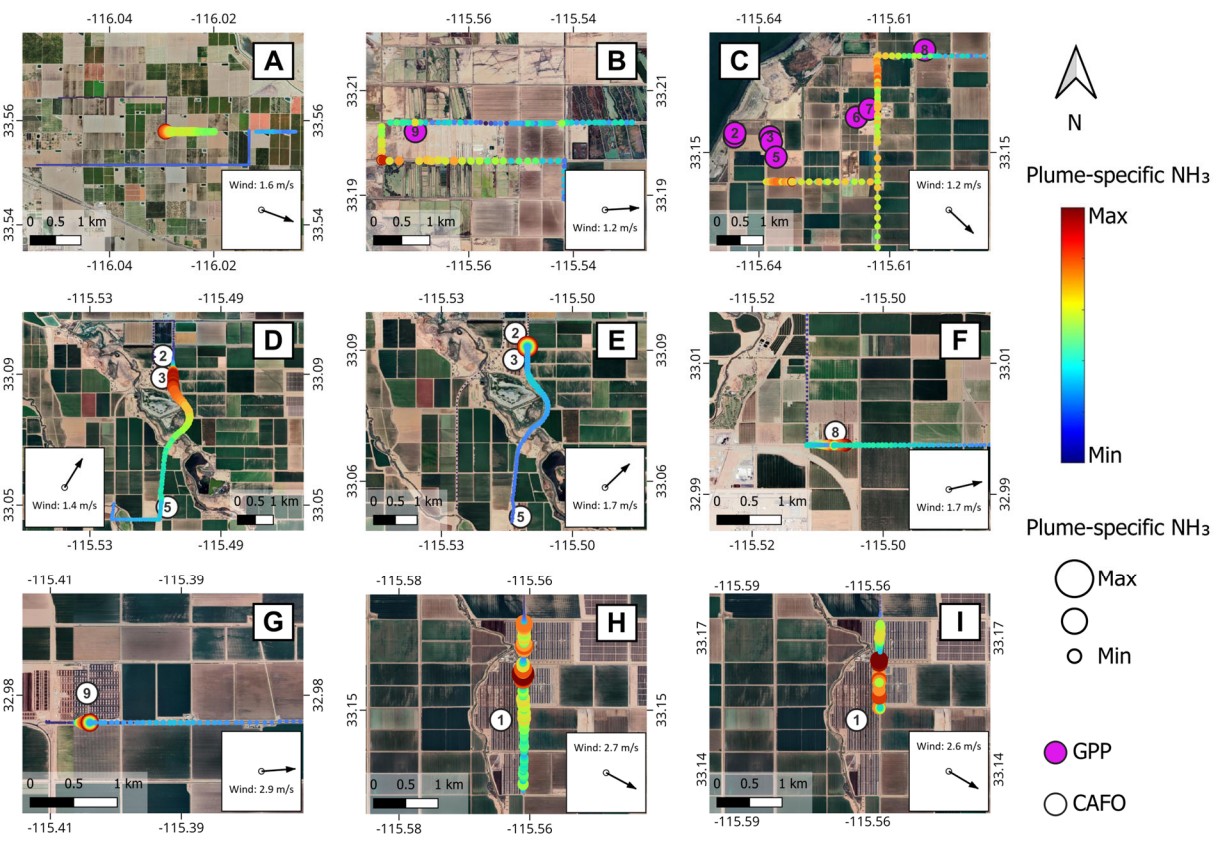


**Figure 9. Maps of individual ground-level NH₃ plumes (A to I), as defined in Figure 8, measured by the mobile platform. The color bar and marker sizes represent the plume specific NH₃ range, where blue and red correspond to the minimum and maximum NH₃ values within each panel, respectively. Because scaling is performed separately for each plume, colors and marker sizes highlight spatial patterns within plumes but are not comparable across panels. Absolute concentrations and cross-plume comparisons are provided in Figures 8 and 10. Average wind speed and direction were calculated from the nearest HRRR grid cells. Base imagery ©2025 Google.**

As shown in Figure 9, nearly all identified plumes, except for plume A, were detected near known NH₃ emission sources. Plume A, the only notable plume detected by the mobile platform in the Mecca study area in March, was observed around some agricultural fields, suggesting NH₃ emissions from fertilizer volatilization. Concurrent airborne NH₃ imagery confirmed that Plume A originated from a diffuse area source linked to the surrounding agricultural fields. Ground observations by the field team also verified that the fields had been freshly fertilized. Ground-level NH₃ concentrations measured near CAFOs were generally higher compared to those recorded downwind of GPPs (Figure 10). The highest NH₃ concentrations were consistently observed near CAFO 1, where the mobile platform recorded substantially elevated NH₃ levels—up to 5,700 ppb—each time it passed downwind of this source.

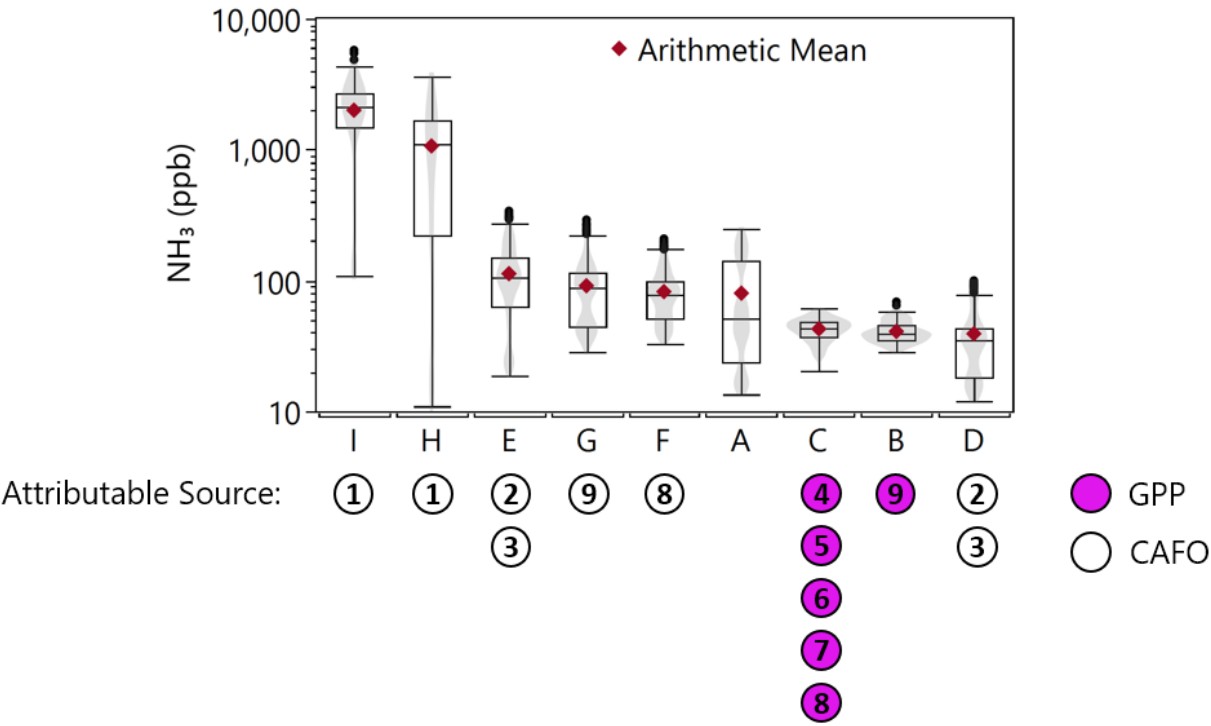

**Figure 10. Box plots of ground-level NH₃ concentrations (ppb) for individual plumes (A to I), as defined in Figure 8, measured by the mobile platform, along with the potential attributable sources. Boxes represent the interquartile range (IQR), with the horizontal line inside each box indicating the median value. Whiskers extend to 1.5 times the IQR, and outliers are values exceeding 1.5 times the IQR. Arithmetic mean values are shown as red diamonds.**

### 3.3. Ground-Based In-Situ Stationary Monitoring

As previously noted, the NH₃ monitor at the Mecca AMS was non-operational in March. Consequently, only the NH₃ data from September 25[th] was used to validate airborne NH₃ measurements on that day, as described in detail

by Tratt et al. (2025). This section presents the longer-term temporal trends and directional patterns of NH$_3$ at this
site, providing insights into its characteristics in the Mecca region. Figure 11 illustrates the monthly averaged NH$_3$
concentrations, showing two notable maxima in winter and late summer, with minima in spring and late fall. Using
April concentrations as a proxy for March, NH$_3$ levels in September, when the second airborne campaign was
conducted, were found to be more than twice those recorded in April. Additionally, a comparison of average NH$_3$
levels across weekdays and weekends (Figure S10) shows no significant differences, suggesting minimal
contributions from sources such as traffic to NH$_3$ emissions in this region.

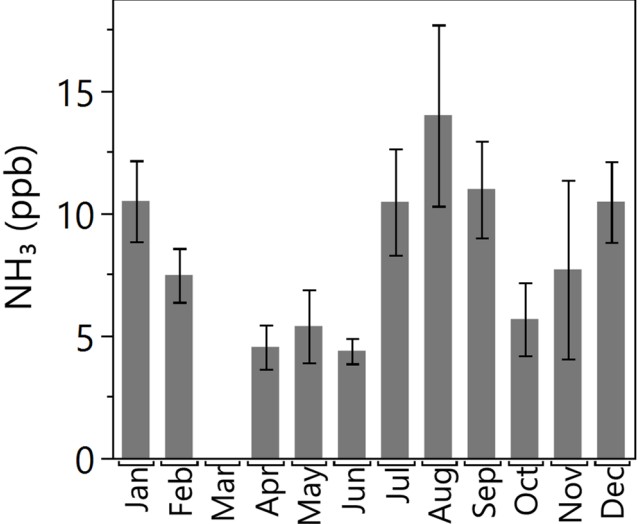


**Figure 11. Monthly averaged ground-level NH$_3$ concentrations (ppb) measured at the Mecca air monitoring site in 2023.**
**Error bars represent confidence intervals. March data is missing due to the NH$_3$ monitor being non-operational during**
**that period.**

The monthly bivariate polar plots in Figure S11 indicate that wind patterns at Mecca are generally balanced between
the northwest and southeast directions. Notably, elevated NH$_3$ levels are most strongly associated with southeasterly
winds, with peak concentrations observed in January, July, August, September, and December. This pattern points to
the likely presence of an NH$_3$ source—or multiple sources—southeast of the Mecca AMS. Given that the Imperial
region, located southeast of Mecca, is known for significantly high NH$_3$ emissions, these emissions are likely the
main contributors to the elevated NH$_3$ concentrations measured at the Mecca AMS.

## 4. Discussion

Pronounced variations in NH$_3$ levels were observed over space and time within the study areas, and the tri-platform
approach provided complementary perspectives in capturing these spatiotemporal trends. Average NH$_3$ levels
measured by the airborne platform were found to be over eight times higher in Imperial compared to Mecca during
the March campaign, and approximately 2.5 times higher during the September campaign. Similarly, mobile
monitoring revealed nearly ten times higher average NH$_3$ levels in Imperial compared to Mecca. Directional analysis
at the Mecca AMS indicated that emissions from Imperial likely contribute to elevated NH$_3$ levels in Mecca.
Seasonally, the average NH₃ column densities over Mecca were approximately 3.5 times higher in September than in
March, while average levels in Imperial remained comparable across both seasons. The seasonal trend observed by
the airborne imaging instrument in Mecca aligns closely with stationary measurements from the Mecca AMS, which
recorded NH₃ levels over 2.5 times higher in September compared to April (used as a proxy for March, as the NH₃
monitor was non-operational).
The emission dynamics between the two study areas were notably distinct. Mecca exhibited considerably lower NH₃
emissions, with the most prominent point source identified being a waste recycling facility. In contrast, Imperial
presented a densely packed array of discrete sources, including several fumaroles, GPPs, and CAFOs. Among them,
CAFO facilities proved to be significant contributors, with larger cattle populations correlating with higher NH₃
emissions. Due to the dispersed layout of CAFO facilities, their plumes were found to be more expansive compared
to those of GPPs, which were narrower and more isolated, predominantly concentrated around cooling towers.
However, more accurate and quantitative assessment of emissions from different sources requires emission flux
calculations, which, as mentioned above, is the focus of another publication from this field campaign.
The measurements presented here reinforce and expand understanding from previous research on atmospheric
ammonia using airborne, satellite, and modeling approaches. For example, Kuai et al. (2019) utilized airborne
HyTES measurements with 2 m spatial resolution in the California San Joaquin Valley to detect NH₃ enhancements
from cattle feedlots, power plants, and smoldering fires, reporting concentrations up to five times higher over
feedlots compared to power plants—findings that closely align with the plume magnitudes and spatial variability
observed in this study. Wang et al. (2021) oversampled IASI satellite data over the contiguous U.S. (CONUS) from
2008 to 2017 to produce ~2-km resolution monthly NH₃ maps, consistently identifying the Imperial Valley as a
hotspot with higher levels in September compared to March, consistent with our findings. Similarly, Pleim et al.
(2019) employed the CMAQ model coupled with the EPIC agricultural model over CONUS to simulate
bidirectional ammonia fluxes, highlighting the significant influence of agricultural fertilizer and livestock emissions
in California's Central and Imperial Valleys.
NH₃ emissions from CAFOs are an emerging concern, particularly with the growing number of these facilities
across the U.S. and the limited understanding of their emissions (Burns et al., 2023; Epps et al., 2025). This gap is
primarily due to the lack of reliable data on the number, size, and precise locations of CAFOs. Consequently, a
significant portion of CAFOs is missing from emission inventories. In fact, a review of the EPA's 2021 National
Emissions Inventory (https://www.epa.gov/air-emissions-inventories/national-emissions-inventory-nei, accessed
January 4, 2025) revealed no records of any CAFOs in our study areas, including CAFO 1, which, according to
CARB's CADD, is the largest CAFO in California, and our measurements revealed substantial emissions from this
facility.
High-spatial resolution, wide-area airborne LWIR imaging provided unprecedented insights into the identification
and quantification of NH₃ sources. The wide-area coverage enabled the survey of relatively large regions within a
short time, capturing all known sources and identifying previously unknown/unreported ones. Simultaneously, the
high-spatial resolution mapping at 2.1 m facilitated the detection of hotspots and the tracing of emission plumes
back to their release points, including isolated and relatively weak $NH_3$ sources such as fumaroles. In contrast,
existing satellite instruments capable of measuring atmospheric $NH_3$ (e.g., CrIS, IASI, AIRS) offer spatial
resolutions of 15–50 km, which are too coarse to resolve localized features. To illustrate this, a hypothetical 15 km x
15 km grid, representative of the finest satellite resolution currently available, was overlaid on an $NH_3$ column
density map from this study (Figure S12). As shown, nearly the entire cluster of geothermal power plants, all
fumaroles, and CAFO 1, the largest in the study area, can fall within a single satellite pixel, masking the spatial
variability captured by the airborne measurements. Additionally, airborne measurement allowed for monitoring areas
that are difficult to access with traditional ground-based or in-situ airborne methods. Another important advantage of
airborne remote sensing, which provides column data, is its ability to capture entire plumes. Surface or airborne in-
situ measurements, by contrast, may miss these plumes due to underflying, overflying, or underdriving the plumes.
Therefore, airborne remote sensing is an ideal tool for air quality managers and regulators for surveying a wide
target area to identify hotspots and guide their subsequent longer-term in-situ monitoring.
While LWIR hyperspectral imaging is highly effective in detecting, identifying, and tracking $NH_3$ emissions, its
sensitivity relies heavily on thermal contrast (Buckland et al., 2017; Hulley et al., 2016; Kuai et al., 2019). When
this contrast is small, as is often the case during early-morning hours before the ground has warmed, the sensor's
ability to detect low-level $NH_3$ diminishes. This effect was particularly evident in Mecca, where background $NH_3$
levels were already significantly lower than those in Imperial. Conversely, afternoon flights benefited from
increased solar heating, which raised $\Delta T$ and improved detection sensitivity, allowing the instrument to capture $NH_3$
signals more effectively, even at relatively low ambient concentrations. For example, on September 25[th], the
stationary ground monitor at the Mecca AMS recorded similar $NH_3$ ambient concentrations during both morning and
afternoon flight times (around 4 ppb). However, as shown in Figure 4, the airborne measurements detected much
more widespread $NH_3$ signals in the afternoon. This difference was due to the higher average $\Delta T$ in the afternoon
(8.5 °C) compared to the morning (4 °C), which enabled the LWIR instrument to resolve $NH_3$ signatures more
effectively. More in-depth discussion about the sensitivity of the Mako retrievals in this study is provided in Tratt et
al. (2025).
Airborne remote sensing measurements inherently provide only "snapshots" of advected emissions within the
specific times and locations where data are collected. However, repeated measurements can offer insights into
typical conditions. Both JPL's HyTES and The Aerospace Corporation's Mako instruments have flown extensively
across California and selected regions worldwide, consistently detecting ubiquitous $NH_3$ emissions in urban and
rural areas (Buckland et al., 2017; Hulley et al., 2016; Kuai et al., 2019). Yet, no systematic effort has been made to
quantify these emissions. Extensive, long-term historical datasets from these instruments, combined with available
$NH_3$ retrieval algorithms with a good understanding of their uncertainties, present a unique opportunity to identify
and characterize $NH_3$ sources on a much broader scale.
A more cost-effective alternative for achieving broader area coverage and routine observations is satellite-borne
remote sensing. However, existing satellite LWIR instruments capable of measuring $NH_3$ lack the spatial resolution
needed to discern emissions at the local scale, especially in areas with dense sources, such as the Imperial region
studied here. Advances in hyperspectral infrared imaging spectrometers and high-resolution detector technologies
now enable $NH_3$ mapping from space-based instruments with higher spatial resolution. For instance, the European
Space Agency recently proposed Nitrosat, a satellite mission concept for high-resolution $NH_3$ monitoring, with
swaths over 80 km and a spatial resolution of 500 m (Noppen et al., 2023). Although not yet selected for further
development, Nitrosat represents significant progress in this area. Additionally, emerging High Operating
Temperature (HOT) Barrier Infrared Detectors (HOT BIRDs) technology (Wright et al., 2020) offers the potential
for lightweight, multi-band infrared detectors with minimal cooling requirements. These detectors can provide
space-based LWIR imaging at the spatial resolutions needed to monitor localized $NH_3$ sources, making them
promising for future advancements in $NH_3$ monitoring.

## 5.    Conclusions

$NH_3$ is increasingly recognized as an emerging pollutant of concern due to its role in secondary $PM_{2.5}$ formation and
nitrogen deposition, both of which have significant environmental and public health impacts. Accurate
characterization of $NH_3$ emissions is essential for improving emission inventories and developing more effective air
quality management strategies. This study demonstrates the integration of airborne LWIR hyperspectral imaging
with ground-based stationary and mobile monitoring for mapping and characterizing $NH_3$ emissions at hyperlocal
scales across two distinct seasons in the Salton Sea region. Seasonal differences were notable over Mecca, with $NH_3$
column densities approximately 3.5 times higher in September than in March, while levels in Imperial remained
relatively comparable across seasons. The high spatial resolution of the LWIR imager revealed pronounced spatial
variability in $NH_3$ levels, with Imperial exhibiting average $NH_3$ column densities 2.5 to 8 times higher than Mecca—
a disparity primarily driven by sources such as CAFOs, geothermal power plants, fumaroles, and intensive
agricultural activities. Notably, the ability to detect previously unreported emission sources, especially in densely
populated agricultural areas, underscores the potential of this approach to improve the identification and
characterization of $NH_3$ emission sources. These findings provide key insights into local-scale emission patterns that
are often missed by traditional ground-based or satellite measurements, highlighting the need for more routine,
wide-scale airborne campaigns and the development of new satellite missions with enhanced resolution to fill
critical data gaps.

**Data availability**

Data from this study can be made available upon request.

**Author Contributions**

Conceptualization: SH, DMT, and OVK; methodology: DMT, CSC, SH, and OVK; flight planning and airborne data
acquisition: ERK; airborne data processing: CSC and KNB; data visualization and plume dispersion analysis: CSC;
ground mobile data acquisition and processing: YM; meteorological model data: MA; stationary measurements: PP
and MS; writing—original draft: SH; writing—review and editing: all authors. All authors have read and approved
the final version of the manuscript.

**Competing interests**

The authors declare that they have no conflict of interest.

**Acknowledgments**

This research was conducted at the Jet Propulsion Laboratory, California Institute of Technology, under a contract with NASA, with support from The Aerospace Corporation's Independent Research and Development program, and Research and Technology Collaborations Office, and the University of California, Riverside. We would like to thank Drs. Toshihiro Kuwayama and Morteza Amini for their valuable input and thoughtful perspectives, which significantly enriched this work. The views and conclusions expressed in this paper are those of the authors and do not necessarily reflect the official policies or positions of their affiliated institutions.

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
