# Peer review of "Region: Insights from Airborne Longwave-Infrared"

_EGUsphere, 2025_

## Author Comment (AC2)

**Reviewer 1:**

*This manuscript integrates airborne, ground-based, and mobile measurements to map NH3 emissions in two source regions in California. The paper is well written and the structure is clear. The results are important for improving the ammonia emission inventories. I only have minor comments as follows:*

*Presentation issues:*

*Section 3.1: Please quantitatively summarize the overall performance of airborne measurements to identify emissions sources, e.g., how many CAFOs/GPPs could be identified.*

**Response**: To address the reviewer's comment, the following paragraph has been added to Section 3.1:

**Line 329-333:** "Overall, the airborne measurements demonstrated a strong capability in identifying $NH_3$ emission sources. Quantitatively, the surveys detected $NH_3$ emissions from all known fumaroles, all 12 GPPs, and all 13 known CAFOs within the study area. Additionally, the high spatial resolution enabled the identification of at least three previously uninventoried point emission sources, appearing as CAFOs based on satellite imagery but not listed in existing databases, highlighting the method's ability to discover undocumented emitters."

*Figure 8: I am not sure if plume A to I were chosen based on the time series or emission sources - please clarify.*

**Response**: Plumes A to I were identified based on distinct peaks in the time series plot (Figure 8). This has been clarified in the revised manuscript:

**Line 343-346:** "To pinpoint the locations and potential emission sources, individual plumes were first identified based on distinct peaks in the time series plot (Figure 8), and their corresponding spatial locations were then mapped in Figure 9."

*Figure 9: There is no marked source in plume A. I suggest switching the colorbar to log scale and using the size of the marker to denote cattle population for each CAFO facility. Also, please add latitudes, longitudes, and wind direction for each panel.*

**Response**: As noted in the manuscript, Plume A originated from a diffuse area source associated with surrounding agricultural fields rather than a discrete point source such as a CAFO. This explains the absence of a marked source near Plume A.

We have incorporated latitude and longitude coordinates and wind direction indicators for each panel in Figure 9, as recommended (a copy of the updated figure is provided below as

well). However, we chose to retain a linear color scale for $NH_3$ concentration and sized the mobile monitoring markers based on measured $NH_3$ values. Using marker size to represent cattle population made the figure visually cluttered; readers can refer to Figure S3 for cattle population data for each CAFO.

[Figure]

**Figure 9. Maps of individual ground-level $NH_3$ plumes (A to I), as defined in Figure 8, measured by the mobile platform. The color scale and marker sizes represent relative $NH_3$ values, with absolute values provided in Figures 8 and 10. Locations of potential known sources near each identified plume are indicated using various markers on the map. Average wind speed and direction were calculated from the nearest HRRR grid cells. Base imagery ©2025 Google.**

*Discussions: It would be helpful to compare the resolution of satellite maps versus airborne measurements in this study to highlight the advantage of this study.*

**Response**: The following discussion and figure have been added to the revised manuscript to address the reviewer's comment:

**Line 437-442**: "Existing satellite instruments capable of measuring atmospheric $NH_3$ (e.g., CrIS, IASI, AIRS) offer spatial resolutions of 15–50 km, which are too coarse to resolve localized features. To illustrate this, a hypothetical 15 km x 15 km grid, representative of the finest satellite resolution currently available, was overlaid on an $NH_3$ column density map from this study (Figure S13). As shown, nearly the entire cluster of geothermal power plants, all fumaroles, and CAFO 1, the largest in the study area, can fall within a single satellite pixel, masking the spatial variability captured by the airborne measurements."

[Figure]

**Figure S13. $NH_3$ column density (ppm-m) map over the Imperial study area on 28 March 2023, overlaid with a hypothetical 15 km x 15 km satellite grid. Base imagery ©2025 Google.**

*Conclusions: I recommend stating the conclusions more quantitatively.*

**Response**: To address the reviewers' comment, the following sentences have been updated or added to the Conclusions section.

**Line 485-492**: "This study demonstrates the integration of airborne LWIR hyperspectral imaging with ground-based stationary and mobile monitoring for mapping and characterizing $NH_3$ emissions at hyperlocal scales across two distinct seasons in the Salton Sea region. Seasonal differences were notable over Mecca, with $NH_3$ column

densities approximately 3.5 times higher in September than in March, while levels in Imperial remained relatively comparable across seasons. The high spatial resolution of the LWIR imager revealed pronounced spatial variability in $NH_3$ levels, with Imperial exhibiting average $NH_3$ column densities 2.5 to 8 times higher than Mecca—a disparity primarily driven by sources such as CAFOs, geothermal power plants, fumaroles, and intensive agricultural activities."

*Minor comments:*

*Line 35: add citation*

**Response**: The following citations have been added to the referenced sentence:

**Line 35-36**: "Ammonia ($NH_3$) is a critical component of the atmospheric nitrogen cycle and an important precursor for the formation of fine particulate matter ($PM_{2.5}$) (Tang et al., 2025; Wyer et al., 2022)."

*Line 38: Please cite commonly used ammonia emission inventories here besides Zeng et al., 2018*

**Response**: Additional citations have been added to the referenced sentence:

**Line 36-38**: "It is primarily emitted from agricultural activities, particularly livestock waste and the application of fertilizers, but also from combustion sources such as vehicles and biomass burning (Beaudor et al., 2025; Vira et al., 2022; Zeng et al., 2018)."

*Line 57: needs citation for the first sentence of the paragraph*

**Response**: A citation has been added to the referenced sentence.

**Line 60-61**: "Despite its importance, measuring and characterizing $NH_3$ emissions are challenging due in part to its short atmospheric lifetime, high spatial heterogeneity, reactivity, and solubility (Nair and Yu, 2020)."

*Line 160: Please briefly discuss the detection limit and validation results here instead of referring to the literature*

**Response**: The following sentences have been added to section 2.4.

**Line 164-167**: "The minimum detectable concentration (MDC) of ground-level $NH_3$ depends on thermal contrast and boundary layer height, and ranged from 3 to 33 ppbv over Mecca in September under varying atmospheric conditions. Validation against groundbased in situ measurements showed good agreement, with differences ranging from 16% to 37% under stable and well-mixed conditions. Detailed descriptions of the retrieval process, detection sensitivity, and validation results for the airborne component of this study are provided in Tratt et al. (2025)."

**Reviewer 2:**

*This study presents measurements of atmospheric ammonia using ground-based and air-borne measurements at two different US cites. The study is more suitable for "Atmospheric Measurement Techniques" than ACP, but can be considered as a "Measurement Report" in ACP. The MS is relatively well-written and the measurements are valuable, but a substantial revision is necessary before it can be accepted for a publication.*

**Response**: We thank the reviewer for the feedback. This study employs well-established measurement techniques and does not introduce new method development, calibration, or validation approaches—areas that align more closely with the scope of Atmospheric Measurement Techniques. Instead, it applies these proven methods to the mapping and characterization of atmospheric ammonia, an emerging pollutant of concern. This application-focused investigation advances understanding of the atmospheric state and behavior of ammonia, which fits directly within the scope of Atmospheric Chemistry and Physics. While the study is geographically focused, its findings extend beyond the local scale by demonstrating the capability of emerging remote sensing technologies to detect and quantify $NH_3$ emissions at hyperlocal resolution in one of the major ammonia hotspots in the US, characterized by a dense variety of sources. This work addresses a critical gap in current atmospheric models and emission inventories, which often underestimate ammonia sources, particularly from CAFOs and geothermal activity. Therefore, we strongly believe this manuscript provides novel scientific insights warranting publication as a research article in Atmospheric Chemistry and Physics.

*Major points:*

1. *Introduction has to be enhanced with recent regional and global studies on atmospheric ammonia and policies related to it. The authors now list only two such studies, which are not enough for an Introduction to any study such studies. One example for a recent study is https://doi.org/10.1016/j.jclepro.2023.140424 and has to be cited.*

**Response**: Introduction has been enhanced by citing more studies, including the one suggested by the reviewer.

**Line 43-45**: "Recent studies have reported increasing trends in atmospheric $NH_3$ levels at global and regional scales, driven largely by agricultural activities, industrial processes, and changing climatic conditions (Kuttippurath et al., 2024; Ma et al., 2025; Van Damme et al., 2021; Wang et al., 2023; Yu et al., 2018)."

**Line 49-50**: "Also, during the 2020 COVID-19 lockdown, Kuttippurath et al. (2024) found that unlike most pollutants which declined, $NH_3$ levels rose, largely due to sustained or increased agricultural activity."

2. *Please compare your measurements with other available satellite or model results. Although the unit of NH₃ is different, you could still compare the general features.*

**Response**: To address the reviewers' comment, the following paragraph has been added to the Discussion section.

**Line 414-423**: "The measurements presented here reinforce and expand understanding from previous research on atmospheric ammonia using airborne, satellite, and modeling approaches. For example, Kuai et al. (2019) utilized airborne HyTES measurements with 2 m spatial resolution in the California San Joaquin Valley to detect $NH_3$ enhancements from cattle feedlots, power plants, and smoldering fires, reporting concentrations up to five times higher over feedlots compared to power plants—findings that closely align with the plume magnitudes and spatial variability observed in this study. Wang et al. (2021) oversampled IASI satellite data over the contiguous U.S. (CONUS) from 2008 to 2017 to produce ~2-km resolution monthly $NH_3$ maps, consistently identifying the Imperial Valley as a hotspot with higher levels in September compared to March, consistent with our findings. Similarly, Pleim et al. (2019) employed the CMAQ model coupled with the EPIC agricultural model over CONUS to simulate bidirectional ammonia fluxes, highlighting the significant influence of agricultural fertilizer and livestock emissions in California's Central and Imperial Valleys."

3. *What are the reasons for the diurnal variations of atmospheric ammonia (Figure 3)? However, this difference is not observed in Figure 2?*

**Response**: The reasons for the diurnal variations of atmospheric ammonia (as shown in Figure 3) are already discussed in the paper (lines 240–245 of the original manuscript). Briefly, the afternoon increase, which was more pronounced over Mecca, is likely due to higher temperatures enhancing $NH_3$ volatilization from soil and plants, especially where fertilizers were recently applied. Additionally, higher thermal contrast during afternoon flights improves the Mako sensor's detection sensitivity, allowing better identification of

NH$_3$ even at low concentrations. Figure 2, which the reviewer refers to, shows the timing of the airborne measurements and does not illustrate any measurement data.

4. *Why the meteorological data are taken from a model output? No AWS or measurements are available? How good are these model results? What is the horizontal resolution of the model? Did you interpolate the model results to the flight tracks?*

**Response**: As detailed in Tratt et al. (2025), the retrieval algorithm requires near-surface air temperature to compute thermal contrast. However, only a few fixed meteorological stations were operating in the region. Given the large extent of the study area (~12,000 km$^2$), these point measurements could not capture the full spatial variability in near-surface air temperature. We therefore used the HRRR model, which provides gridded data at 3 km resolution and 15-minute intervals. The figure below compares hourly temperature, wind speed, and wind direction from in-situ measurements at the Mecca air monitoring site (AMS) with 15-minutes values from the nearest HRRR grid point during the second flight day (i.e., 25 September, 2023). As shown, HRRR's temperature and wind direction closely match the in-situ data, while wind speed follows similar temporal trends, albeit with occasional deviations in magnitude. It should be noted that while the temperature data were used in the NH$_3$ retrieval, wind data were only used qualitatively in this paper to assess general wind patterns across different parts of the study region.

[Figure]

Comparison of HRRR model and in-situ meteorological data at Mecca AMS

5. *What are the reason for the March and September NH3 differences (L380-390)?*

**Response**: As discussed in the paper, the differences in $NH_3$ levels between March and September are due to higher temperatures in September that enhance $NH_3$ volatilization from soil and plant surfaces, increased thermal contrast during September flights that improves the LWIR sensor's sensitivity, and seasonal variability in emission sources.

6. *Need a discussion of the limitation of the analysis and uncertainty of the retrievals.*

**Response**: The limitations of the study and the uncertainty/sensitivity of the retrievals are discussed extensively in lines 419–439 of the original manuscript. Additionally, information on the detection limits and validation of the airborne measurements has been added to lines 164-167 of the revised manuscript in response to Reviewer 1's comments.

*Minor issues:*

*L60: and AIRS*

**Response**: AIRS is already noted on line 60 of the original manuscript.

*L284-285: What is the scale that you are talking about? Mention this here*

**Response**: This sentence has been revised for clarity:

[revised manuscript text omitted]

---

## Author Response (AR2)

**Reviewer 1:**

*Please clarify how the relative NH3 was calculated in Figure 9. Did you normalize the data by a reference value (median or mean)? If so, denote the color for the reference value on the colorbar.*

**Response**: Thank you for your feedback. To clarify, in Figure 9, the color bar and marker sizes represent the plume-specific $NH_3$ range, where blue and red correspond to the minimum and maximum $NH_3$ values within each panel, respectively. Since the value ranges differ across plumes, showing nine separate color bars and maker size legend would clutter the figure. In the revised version, we have relabeled the color bar and marker size legend as "Plume-specific $NH_3$" and changed the labels "Low" and "High" to "Min" and "Max", respectively. The caption has also been expanded to explicitly state that colors and marker sizes are not comparable across panels, and that absolute concentrations are provided in Figures 8 and 10.

[Figure]

**Figure 9. Maps of individual ground-level $NH_3$ plumes (A to I), as defined in Figure 8, measured by the mobile platform. The color bar and marker sizes represent the plume-specific $NH_3$ range, where blue and red correspond to the minimum and**

maximum $NH_3$ values within each panel, respectively. Because scaling is performed separately for each plume, colors and marker sizes highlight spatial patterns within plumes but are not comparable across panels. Absolute concentrations and cross-plume comparisons are provided in Figures 8 and 10. Average wind speed and direction were calculated from the nearest HRRR grid cells. Base imagery ©2025 Google.